# Contribution of Risk and Resilience Factors to Suicidality among Mental Health-Help-Seeking Adolescent Outpatients: A Cross-Sectional Study

**DOI:** 10.3390/jcm12051974

**Published:** 2023-03-02

**Authors:** Tal Shilton, Nimrod Hertz-Palmor, Noam Matalon, Shachar Shani, Idit Dekel, Doron Gothelf, Ran Barzilay

**Affiliations:** 1Child Adolescent Psychiatry Division, Sheba Medical Centre, Ramat Gan 52621, Israel; 2Sackler School of Medicine, Tel Aviv University, Tel Aviv 69978, Israel; 3Medical Research Council Cognition and Brain Sciences Unit, University of Cambridge, Cambridge CB2 7EF, UK; 4Sagol School of Neuroscience, Tel Aviv University, Tel Aviv 69978, Israel; 5Lifespan Brain Institute, Children’s Hospital of Philadelphia (CHOP) and Penn Medicine, Philadelphia, PA 19104, USA; 6Department of Psychiatry, Perelman School of Medicine, University of Pennsylvania, Philadelphia, PA 19104, USA; 7Department of Child and Adolescent Psychiatry and Behavioral Sciences, Children’s Hospital of Philadelphia, Philadelphia, PA 19104, USA

**Keywords:** peer victimization, suicidality, resilience, child and adolescent psychiatry

## Abstract

Background: Peer victimization is an established risk factor for youth suicidal thoughts and behavior (suicidality), yet most peer-victimized youth are not suicidal. More data are needed pertaining to factors that confer resilience to youth suicidality. Aim: To identify resilience factors for youth suicidality in a sample of N = 104 (Mean age 13.5 years, 56% female) outpatient mental health help-seeking adolescents. Methods: Participants completed self-report questionnaires on their first outpatient visit, including the Ask Suicide-Screening Questions, a battery of risk (peer victimization and negative life events) and resilience (self-reliance, emotion regulation, close relationships and neighborhood) measures. Results: 36.5% of participants screened positive for suicidality. Peer victimization was positively associated with suicidality (odds ratio [OR] = 3.84, 95% confidence interval [95% CI] 1.95–8.62, *p* < 0.001), while an overall multi-dimensional measure of resilience factors was inversely associated with suicidality (OR, 95% CI = 0.28, 0.11–0.59, *p* = 0.002). Nevertheless, high peer victimization was found to be associated with a greater chance of suicidality across all levels of resilience (marked by non-significant peer victimization by resilience interaction, *p* = 0.112). Conclusions: This study provides evidence for the protective association of resilience factors and suicidality in a psychiatric outpatient population. The findings may suggest that interventions that enhance resilience factors may mitigate suicidality risk.

## 1. Introduction

Suicide is the second leading cause of death among individuals between the ages of 10 and 14 and the third leading cause of death among individuals between the ages of 15 and 24 [1], underscoring the need for early identification and interventions to minimize youth suicide risk. Youth suicidal thoughts and behavior (suicidality) is a multifaceted phenomenon with a complex interplay between internal (e.g., psychopathology, neurocognitive) and external (e.g., adversity, supportive environments) risk and resilience factors [2], thus making research on suicide prevention highly challenging [3].

Peer victimization is an established risk factor for youth suicidal ideation [4] and suicide attempts [5,6]. The association between peer victimization and suicidality seems to be a global phenomenon, as it can be found in both high- and low-income regions [5,7]. Studies have found that the association between peer victimization and suicidality is mediated by depression, low self-esteem, hopelessness, loneliness, self-blame, and exacerbation of an adverse family environment [8,9,10,11].

Although research has shown that peer victimization experiences have a negative effect on mental health, not all victims experience the same outcomes. Hence, there has been a growing interest over the past few years regarding the role of factors that confer resilience (i.e., resilience factors) in youth who experience victimization [12,13,14,15,16]. Resilience is considered a dynamic, multidimensional process through which youths adjust to stressful events, and it is shaped by multiple factors including previous adverse experiences, external support, and individual traits [17,18].

To date, data on resilience factors in youth suicidality are relatively scarce, despite their potential role in mitigating suicide risk [19]. To our knowledge, the only study that has described the relationship among peer victimization, resilience factors and suicidality in youth included a cohort of community-based adolescents and focused on individual-level traits such as sociability, communication skills and self-esteem [20]. More data are needed on multi-level (e.g., environmental, familial and interpersonal) resilience factors for youth suicidality [21,22,23,24], especially among clinical mental health-help-seeking population.

In the current study, we investigated associations of resilience factors with suicidality in a clinical adolescent population ascertained from a psychiatric outpatient clinic. We performed a multidimensional assessment of risk and resilience factors using a battery that probed intrapersonal, interpersonal, and social environment characteristics, including adverse family events and peer victimization. In addition, we evaluated patients’ suicidality risk, anxiety, and depressive symptoms. We hypothesized that (1) peer victimization will be associated with suicidality risk; (2) specific resilience factors such as emotion regulation and low family conflict will be associated with less suicidality risk; and (3) resilience factors will moderate the association between peer victimization and suicidality.

## 2. Materials and Methods

### 2.1. Study Sample

We conducted a cross-sectional study of adolescents treated in the Child and Adolescent Psychiatry Outpatient Clinic of a major hospital in central Israel (Sheba Medical Center). A power analysis using G*Power 3.1.9.4 [25] showed that with strong hypothesized effect sizes of odds ratio > 2 (or 0.48 for protective effect), a sample of N = 100 would be required to observe effects with an α criterion < 0.05 with 80% statistical power. A total of 127 adolescents were invited to complete the assessment battery while waiting for their first appointment at the clinic, during a six-month period between 11 June and 7 December 2020. Of this sample, we excluded patients with intellectual disability (n = 2), autism spectrum disorder (n = 1), or psychotic spectrum disorder (n = 1). Ten patients chose not to participate in the study, and 9 (7%) had incomplete data on suicidality, psychopathology or risk and resilience factors, and were therefore excluded from analyses. The final sample included 104 patients. The study was conducted according to the guidelines of the Declaration of Helsinki and approved by the Institutional Review Board at Sheba Medical Center (7212-20-SMC). Informed assent/consent was obtained from the patients and their parents.

### 2.2. Measures

Self-report questionnaires were completed by the patients through a secured digital platform (REDCap). The questionnaires were self-administered, except for cases in which patients experienced difficulties reading or understanding the questions, in which case a member of the research team would read items out loud and provide instructions. Team members were not aware of the patient’s responses, as they only read the questions and enabled patients to click on the relevant answer independently. Parents reported sociodemographic characteristics including age, sex, religion, and parent education. Since our study was conducted in the midst of the COVID-19 pandemic, we collected data on parents’ income and whether the parent was recently laid off or put on unpaid leave due to COVID-19, to address pandemic-specific socioeconomic factors that may impact mental health on the familial level [26,27,28]. Parents were requested to report their income on a 5-point Likert scale, ranging from a lot below average to a lot above average. Categories were accompanied with information about the average wages in Israel at the time, to inform parents on their response. Since only 2 parents reported that their salary was a lot above average, we collapsed the two lowest and two highest categories to create a 3-point Likert scale (below average income, average income and above average income). Study participants also completed the following measures:

Suicidality risk was assessed using the Ask Suicide-Screening Questions (ASQ) toolkit [29], a 4-item nonproprietary suicide-risk screening instrument. The ASQ items inquire about wishes to be dead, feelings of oneself or others being better off if one was dead, thoughts of killing oneself and previous attempts at killing oneself. If a patient’s response is “no” to all 4 items, this is considered a negative screen and no further questions are asked unless clinical judgement overrides the screening result. If a patient responds “yes” to any item or refuses to answer, the screen is considered “positive”, and a fifth question is asked to determine acuity (“Are you having thoughts of killing yourself right now?”) [30]. The tool has been validated for use in pediatric inpatient and outpatient settings for ages 10 to 21, with sensitivity values ≥ 95% and specificity values ≥ 87% [28]. In our current study, all patients answered all 5 questions and a positive answer to current thoughts of suicide was immediately reported to the child psychiatrist appointed to evaluate the child and to offer interventional strategies.

For the current analyses, ASQ scores were collapsed into binary values: youth who endorsed at least one item of the ASQ were considered positive for suicidal risk assessment (coded 1), while youth who did not endorse any ASQ item were considered negative for suicidal risk assessment (coded 0).

Risk and Resilience factors were assessed using self-report of the risk and resilience battery (R&R Battery). This battery was previously developed and administered in English [31,32] and was translated to Hebrew by bilingual speakers using translation and back translation. The R&R Battery was administered by clinically trained research team members (authors TS, NH-P, NM and SS) at the outpatient clinic prior to their first appointment with a psychiatrist. The R&R Battery includes 47 self-report items and combines seven intrapersonal, interpersonal, and external/environmental factors, coded on either a 7-point or a 5-point Likert Scale. Risk and resilience factors include: (1) self-reliance (3 items; e.g., can usually find a way out of difficult situations. Cronbach’s α = 0.76), (2) emotion regulation (5 items; e.g., difficulty concentrating or controlling behaviors when upset, limited access to emotion regulation strategies. α = 0.81), (3) positivity and support in close relationships (4 items; e.g., lasting relationship and level of care. α = 0.78), (4) negativity and hostility in close relationships (5 items; e.g., level of arguing. α = 0.83), (5) perceptions of the neighborhood environment (4 items; e.g., perceived level of trust and safety in neighborhood. α = 0.69), (6) peer victimization (12 items; e.g., called names or harassed online. α = 0.89), and (7) negative life events (11 items on personal and family-related stressors; e.g., parental divorce, move, family in trouble with the law. α = 0.67). To maximize interpretability, we re-coded reverse items to represent higher risk for risk subscales (peer victimization, negative life events), and higher resilience for resilience subscales (self-reliance, emotion regulation, positive relationships, lack of negative relationships and neighborhood safety).

For the current analyses, and consistent with our past works using the R&R Battery [33], items were aggregated to create continuous scales. An overall resilience factors score was quantified by summing the items of the following subscales: self-reliance, emotion regulation, positive relationships, lack of negative relationships and neighborhood safety. Alongside the overall resilience factors score, we calculated each subscale score separately to create a continuous subscale for each resilience domain.

Depression screening was conducted using the Patient Health Questionnaire-9 (PHQ-9) [34], a self-report scale developed to assess the defining symptoms of depression (for example: “Feeling down, depressed, or hopeless”). The items are rated on a 4-point Likert-type scale (from 0 = not at all to 3 = nearly every day) and scores ranged from 0 to 21. The internal reliability (Cronbach’s α) in the current sample was 78.

Anxiety screening was conducted using the Generalized Anxiety Disorder 7 (GAD-7) questionnaire [35], a self-report scale developed to assess the defining symptoms of anxiety (for example: “Feeling nervous, anxious or on edge”). The items are rated on a 4-point Likert-type scale (from 0 = not at all to 3 = nearly every day) and scores ranged from 0 to 21. The internal reliability in the current sample was 86.

### 2.3. Data Analysis

#### 2.3.1. Characterization of Suicidal Adolescents and Univariate Comparison

We compared demographics and risk and resilience factors between suicidal youth to their non-suicidal peers using t-tests for continuous factors and χ^2^ tests for discrete variables. Comparisons of risk factors included peer victimization and total negative life events assessed in the R&R Battery. Comparisons of resilience factors included self-reliance, emotion regulation, family factors and the neighborhood measures included in the R&R Battery. We also compared depression and anxiety symptoms assessed in the PHQ-9 and GAD-7, respectively.

#### 2.3.2. Multivariable Models

To assess the combined contribution of risk factors and the overall resilience factors score to suicidality in the sample, we estimated a logistic regression model with suicidality as the dependent variable. The independent variables were two risk factors: peer victimization and total negative life events; and the overall resilience factors score. A separate model also probed for peer victimization based on the overall resilience factors score interaction.

To explore the associations of each resilience domain with suicidality risk, we conducted a post hoc hierarchical multiple logistic regression in two steps. In step I, we introduced age, sex, peer victimization and the five resilience subfactors. In step II, we explored the interactions of each resilience subfactor with peer victimization.

All multivariable models included the following covariates: age (continuous), sex (ref. = male), parents’ income (ordinal, 1–5), parents’ recent layoff (ref. = no layoff). Continuous predictors were standardized to facilitate results interpretation. Due to high collinearity among peer victimization and the general resilience factor, these variables were detrended of shared variance before they were introduced to the model, by regressing them from one another and introducing their standardized residuals to the final model [36]. A similar approach was implemented at post hoc analyses, where peer victimization and the specific resilience factors were detrended before introducing them to the model (each specific factor was also detrended from the variance shared with the rest of the specific resilience factors).

### 2.4. Sensitivity Analyses

To address specificity between risk and resilience factors and suicidality over and above non-suicidality psychopathology, we conducted a sensitivity analysis in which we included PHQ-9 and GAD-7 scores to control for the potential effects of internalizing symptoms on suicidality. Since one of the items in PHQ-9 inquires about suicidal ideation directly (item i.—“thoughts that you would be better off dead or of hurting yourself in some way”), this item was not included as a covariate, and instead all other items were aggregated to formulate “PHQ-8” scores that represent depressive tendencies. In all of our analyses we used a standard α < 0.05 chance of a type I error. Analyses were conducted using the “stats” package (version 4.0.3) in R [37].

## 3. Results

Sample characteristics are detailed in Table 1. Of the 104 adolescents who met inclusion criteria, 38 (36.5%) endorsed at least one ASQ item and were considered positive for suicidal risk assessment; the rest (66 participants, 63.5%) were considered not suicidal. The mean age in the total sample was 13.5 ± 2.1. Participants were mostly female (56.1%), and 39.3% of parents reported their income was average, while 32.7% and 25.2% reported below- and above-average income, respectively. Approximately one out of five participants (18.6%) had parents who were recently laid off or put on unpaid leave as a result of the coronavirus pandemic.

In multivariable analysis, peer victimization was associated with increased odds of suicidality (odds ratio [OR] = 3.84, 95% confidence interval [95% CI] = 1.95–8.62, *p* = *0*.0003), while the total number of negative life events was not (OR = 0.91, 95% CI = 0.48–1.66, *p* = *0*.770). The overall resilience factors score was associated with reduced odds of suicidality (OR = 0.28, 95% CI = 0.11–0.59, *p* = *0*.002). There was no peer victimization based on overall resilience factors score interaction (*p* = 0.112), indicating no evidence for a moderating effect of resilience on the association between peer victimization and suicidality (Table 2). Across the study population, participants with low and high levels of peer victimization demonstrated similar protective association of resilience factors against suicidality (Figure 1). These results remained when accounting for depression and anxiety scores (Appendix A).

To explore the contribution of the specific resilience subfactors to suicidality, we tested their association with suicidality, accounting for the peer victimization risk. We found that all five resilience domains were associated with reduced odds of suicidal risk (Self-reliance: OR = 0.43, 95% CI = 0.22–0.83, *p* = 0.014; Emotion regulation: OR = 0.32, 95% CI= 0.15–0.64, *p* = *0*.002; Positive relationships: OR = 0.32, 95% CI = 0.15–0.64, *p* = 0.002; Lack of negative relationships: OR = 0.36, 95% CI = 0.17–0.72, *p* = 0.005; Neighborhood safety: OR = 0.43, 95% CI = 0.22–0.79, *p* = 0.009). None of the resilience subfactors moderated the association between peer victimization and suicidality (all interactions *p*’s > 0.05, Table 3).

## 4. Discussion

This study provides evidence of the association between risk and resilience factors and suicidality in a psychiatric outpatient youth population, adding to the limited literature on the role of resilience factors in youth suicidality [20]. We found that peer victimization, the main risk factor we included in our assessment, was strongly associated with suicidality (OR~4), as expected based on the established literature on bullying and suicide risk in adolescence [5]. Our multidimensional resilience factors score, comprised of intrapersonal and interpersonal factors and of neighborhood environment, was negatively associated with suicidality even among peer-victimized youth, such that for a 1 standard deviation increase in the resilience score, the odds of suicidality decreased ~3-fold. This finding adds to the existing literature on this resilience factor score’s association with mental health burden in adults [33,38] and in adolescents who are not seeking mental health help [32]. Importantly, the overall resilience factors score had a protective association with suicidality even among youth who are highly victimized, underscoring the potential of enhancing resilience factors among youth at risk for suicidality as part of suicide prevention intervention initiatives.

A few strengths are notable due to their potential clinical and research implications. First, we studied resilience factors in a clinical population of outpatient adolescents at high risk of suicidality based on the 36.5% suicidality rate, which is substantially higher than that expected in the general adolescent population [39,40]. Second, the protective association of the overall resilience factors score with suicidality remained significant even in our sensitivity analyses that accounted for depressive and anxiety symptoms, suggesting that resilience factors may be relevant to mitigate suicide risk specifically, over and above co-existing internalizing symptomatology. Third, a key finding is that even in our high-risk population, all five separate domains of our resilience factors’ evaluation were associated with lower odds of suicidality. Moreover, our results highlight the protective association of these individual domains, as well as the utility of modeling resilience as a multi-level construct when investigating suicidality risk in clinical samples. We have previously described our experience assessing these resilience factors using a battery that represents a multi-level approach to resilience in other populations [31,38,41]. Our findings add to the existing literature knowledge by showing that intrapersonal (self-reliance, emotion regulation), interpersonal (positivity and support in close relationships,) and broader neighborhood environmental context are protective factors among the high-risk adolescent population.

Our results lend further support to the literature suggesting the fostering of resilience to mitigate mental health burden [21,22,42]. Examination of the individual resilience factors suggests that youth with a greater sense of support in close relationships were generally found to have lower suicidal risk. Indeed, family and parental support is a relevant protective factor that has been frequently examined in bullying [43] and suicide [44] research. Victims who lack family support appear to be more vulnerable to suicidal ideation [45], and on the other hand, youth who report strong family connections [42] and those who receive adequate social support at home may be less prone to contemplate suicide even when victimized [46]. In addition, emotion regulation and self-reliance, two intra-personal resilience factors, were associated with lower suicidality risk, contributing to the current knowledge on individual-level resilience factors involved in suicide research.

Nevertheless, high peer victimization was found to be associated with a higher chance of suicidality across all levels of resilience, even among youth with high resilience score. Results highlight the importance of peer victimization as a risk factor, its relevance to the developmental process of children and adolescents [47] and the need for effective peer victimization prevention interventions [48]. Victimization by peers is highly prevalent in adolescence, with rates ranging from 11% to 40% [49], and it has been associated with concurrent suicidal ideation and suicide attempt [4,7]. Hence, future longitudinal studies are needed to provide insight into the contribution of peer victimization and resilience factors to suicide risk over time. Moreover, these studies could potentially target resilience factors in a tailored manner. For example, high-risk youth with low emotional regulation capabilities might benefit from intervention of dialectical behavior therapy that specifically address emotion regulation [50]. In addition, interventions aimed at enhancing resilience factors can be incorporated in focused interventions targeting suicide risk, which are common in Israel [51,52].

Adolescence is a dynamic developmental period characterized by behavioral changes and neural adaptation, which are critical for resilience outcomes [53]. These factors highlight the potential of adolescence as an ideal period for interventions aimed at enhancing resilience [54]. These interventions can theoretically be administered before, during or following an acute stress exposure, such as peer victimization. However, more research is needed to properly assess the efficacy of interventions to foster resilience [55]. Our results support future studies in considering emotion regulation and self-reliance as possible targets for intervention to increase resilience, especially among high-risk victimized adolescents [56,57,58,59]. Other resilience factors, such as positive and negative aspects of close relationships and neighborhood safety, had a negative association with suicidality. This might highlight the importance of social cohesion alongside family and community support [60,61].

This study was conducted in the first year of the COVID-19 pandemic, a unique point in time that may influence the interpretation and generalization of findings. We accounted for some potential confounders that we thought could affect the mental health and family dynamics of patient’s families, including income and job loss due to the pandemic. However, the timing of data collection may affect the patient population who reached out for treatment in such unique circumstances [62,63,64]. For example, these families could have greater resilience factors that allowed them to seek care in difficult circumstances, causing an ascertainment bias for families with greater resilience factors. This timing also may affect youth suicidality risk, though the pandemic’s exact effects on suicidality are still unclear [65].

Our findings should be interpreted taking some limitations into consideration. First, our sample included 104 psychiatric outpatients, and one should be careful regarding generalizing results to other settings. Nonetheless, the fact that we systematically assessed risk and resilience factors and suicidality supports generalizability to an outpatient setting, as our hospital serves a catchment area of ~0.5 million citizens and a diverse population in terms of socioeconomic status. Second, we included youth aged 10–18; hence, our findings may not generalize to suicidality in younger (elementary school age) children. Third, our measure of risk factors did not include other key risk factors such as exposure to screen time, drug abuse, sexual activity or eating disorders, which have been found to be risk factors for suicidality in peer-victimized youth [66,67]. Fourth, although the R&R battery focus on intrapersonal, interpersonal and social environment characteristics, resilience clearly involves certain lifestyle factors, such as physical activity and sleep quality [68,69], which were not assessed in the current study. Future studies should incorporate lifestyle factors when measuring resilience, as they might have a potential role in suicide prevention and intervention strategies [70]. Lastly, this was a cross-sectional study and one cannot infer causality from our work. Future longitudinal studies are needed to clarify the causal relationship between resilience factors and suicidality among youth.

## 5. Conclusions

To conclude, we provide insight regarding the relationship between risk and resilience factors and suicidality in a high-risk adolescent population. Findings underscore the critical role of peer victimization during adolescence as a suicidality risk factor and reveal the multifaceted nature of resilience factors that can have a protective role in mitigating suicidality risk, even among peer victimized youth. Our findings suggest that fostering of resilience factors such as self-reliance and emotion regulation may be beneficial in mitigating suicide risk. Additionally, clinical attention to adolescents’ views on their close relationships and neighborhood safety, two resilience factors that were associated with lower suicidality in our study, may help improve suicidality risk classification and inform preventative interventions such as safety planning. Future studies are needed to prospectively test whether resilience-enhancing interventions can reduce the risk of suicidality over time.

## Figures and Tables

**Figure 1 jcm-12-01974-f001:**
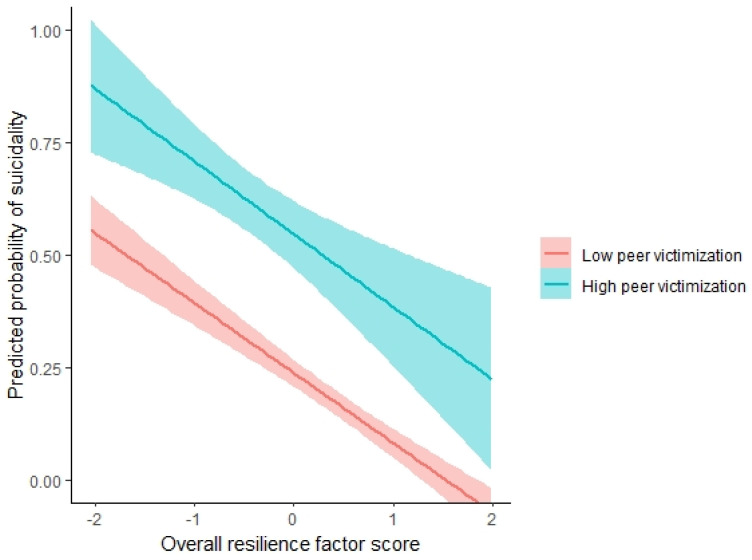
Association among overall resilience factors score (X axis, standardized scores) and predicted chances of suicidality (Y axis, odds of predicted suicidality [operationalized as ASQ score ≥ 1] range between 0–1 with 95% confidence interval). Low peer victimization (red) group = youth in the bottom tertial (0–33%) of peer victimization scores. High peer victimization (blue) group = youth in the top tertial (66.7–100%) of peer victimization scores. Greater overall resilience factors score is associated with lower odds of suicidality independently of peer victimization levels.

**Table 1 jcm-12-01974-t001:** Sociodemographic and clinical characteristics of the study sample.

Characteristics	Entire Sample (N = 104)	Non-Suicidal (n = 66)	Suicidal (n = 38)	*p*
Age, years mean (SD)	13.5 (2.1)	13.23 (2.15)	14.09 (1.84)	0.046
Age, years range	10–18	10–18	10.5–17.7	-
Sex, female, n (%)	60 (56.1%)	33 (50.0%)	26 (68.4%)	07
Family members living in the house	4.2 (1.0)	4.2 (1.0)	4.2 (1.0)	0.95
Religion, n (%)				0.17
Secular	62 (57.9%)	34 (51.5%)	27 (71.1%)	
Religious	36 (33.7%)	26 (39.4%)	8 (21.1%)	
Ultra-orthodox (Haredi)	3 (2.8%)	3 (4.5%)	0 (0.0%)	
Perceived self-health relative to peers (1–5 Likert scale)	3.2 (1.0)	3.3 (1.0)	3.0 (1.0)	0.13
Parents education				0.056
<High school	7 (6.5%)	3 (4.7%)	4 (10.8%)	
High school graduate	41 (38.3%)	27 (42.2%)	13 (35.1%)	
Bachelor’s degree	34 (31.8%)	25 (39.1%)	8 (21.6%)	
Master’s degree or higher	22 (20.6%)	9 (14.1%)	12 (32.4%)	
Parents’ income				0.19
Below average	35 (32.7%)	26 (40.6%)	9 (24.3%)	
Average	42 (39.3%)	25 (39.1%)	16 (43.2%)	
Above average	27 (25.2%)	13 (20.3%)	12 (32.4%)	
Parents’ layoff/unpaid leave	20 (18.6%)	15 (22.7%)	5 (13.2%)	0.23
Risk factors, mean (SD)				
Peer victimization scale, mean (SD)	6.1 (5.4)	4.5 (4.4)	9.0 (6.0)	**<0.0001**
Life events scale, mean (SD)		2.7 (2.1)	2.9 (2.0)	0.75
GAD-7 scale, mean (SD)	8.1 (5.4)	7.0 (5.0)	10.0 (5.6)	**0.011**
PHQ-9 scale, mean (SD)	8.7 (5.3)	7.1 (4.5)	11.6 (5.7)	**<0.0001**
Resilience sub-factor scales, mean (SD)				
Self-reliance	14.4 (4.8)	15.2 (4.6)	13.0 (4.9)	**0.024**
Emotion regulation	16.9 (5.1)	18.0 (4.7)	14.9 (5.2)	**0.002**
Positive relationships	15.9 (3.8)	16.7 (3.5)	14.4 (4.0)	**0.002**
Lack of negative relationships	18.3 (4.7)	19.2 (4.3)	16.6 (4.9)	**0.007**
Neighborhood safety	15.4 (3.3)	16.0 (3.1)	14.4 (3.3)	**0.014**

Significant associations are marked in bold. Abbreviations. N = Total study population size; n = sample size of subsample within study population; SD = standard deviation; GAD-7 = Generalized Anxiety Disorder-7; PHQ-9 = Patient Health Questionnaire-9.

**Table 2 jcm-12-01974-t002:** Multivariate logistic regression with ASQ ≥ 1 as dependent variable.

Factor	OR (95% CI)	*p*
Age ^a^	1.52 (0.92, 2.56)	0.11
Sex ^b^	1.45 (0.46, 4.70)	0.52
Parents’ income ^a^	1.14 (0.66, 2.01)	0.64
Parents layoff during COVID-19 ^b^	0.49 (0.10, 1.94)	0.33
Total number of negative life events ^a^	0.91 (0.48, 1.66)	0.77
Peer victimization ^a^	3.84 (1.95, 8.62)	**0.0003**
Resilience factors score ^a^	0.28 (0.11, 0.59)	**0.002**
Resilience factors score-by-Peer victimization interaction	1.38 (0.84, 2.36)	0.21

Significant associations are marked in bold. ^a^ Standardized scores; ^b^ Binary-values. Abbreviations. ASQ = Ask Suicide-Screening Questions. OR = odds ratio. CI = confidence interval. *p* = *p*-value (significance). COVID-19 = Coronavirus Disease 2019.

**Table 3 jcm-12-01974-t003:** Multiple hierarchical logistic regression with ASQ ≥ 1 as dependent variable.

	STEP I		STEP II	
Factor	OR (95% CI)	*p*	OR (95% CI)	*p*
Peer victimization	3.13 (1.81, 5.91)	**0.0001**	3.64 (2.01, 7.39)	**<0.0001**
Self-reliance	0.43 (0.22, 0.83)	**0.014**	0.43 (0.21, 0.84)	**0.016**
Emotion regulation	0.32 (0.15, 0.64)	**0.002**	0.32 (0.14, 0.65)	**0.003**
Positive relationships	0.32 (0.15, 0.64)	**0.002**	0.31 (0.14, 0.63)	**0.002**
Lack of negative relationships	0.36 (0.17, 0.72)	**0.005**	0.34 (0.16, 0.69)	**0.004**
Neighborhood safety	0.43 (0.22, 0.79)	**0.009**	0.45 (0.22, 0.82)	**0.014**
Peer victimization-by-Self-reliance interaction	-	-	1.70 (0.86, 3.49)	0.13
Peer victimization-by-Emotion regulation interaction	-	-	1.68 (0.91, 3.44)	0.12
Peer victimization-by-Positive relationships interaction	-	-	1.30 (0.61, 2.85)	0.50
Peer victimization-by-Lack of negative relationships interaction	-	-	0.80 (0.34, 1.80)	0.59
Peer victimization-by-Neighborhood safety interaction	-	-	1.43 (0.87, 2.58)	0.19

Significant associations are marked in bold. All odds ratios refer to standardized scores. Abbreviations: ASQ = Ask Suicide-Screening Questions. OR = odds ratio. CI = confidence interval. *p* = *p*-value (significance). COVID-19 = Coronavirus Disease 2019.

## Data Availability

The datasets generated during and/or analyzed during the current study are available from the corresponding author on reasonable request.

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
