# Peer review of "Contribution of Risk and Resilience Factors to Suicidality among Mental Health-Help-Seeking Adolescent Outpatients: A Cross-Sectional Study"

_jcm, 2023, doi:10.3390/jcm12051974_

Round 1

Reviewer 1 Report

Thank you for the opportunity to review "Contribution of Risk and Resilience Factors to Suicidality Among Mental Health Seeking Adolescent Outpatients: A 3 Cross-Sectional Study. The paper addressed a topic of significant importance by examining the relationship between suicidality, peer-victimization, and resilience in youth in a psychiatric outpatient setting. Overall, the manuscript was clear and concise. The introduction included adequate literature that led to specific research questions. The measures and methods were described and replicable, and the Tables and Figures provided a good display of the results. The Discussion reviewed the results and provided convergence with the literature, limitations, and recommendations for future research. Two areas that could be improved include: 1) adding information on the internal consistency of the scales for the resilience measure; 2) in the first paragraph of the Discussion much of the convergence with the literature appears to be limited to the work of co-authors and is missing other relevant studies that are applicable to the findings. I would suggest expanding this section to include other research that can aid in the clinical implications of the findings as well as guiding future experimental research. 

Author Response

Reviewer #1

Comment:

Thank you for the opportunity to review "Contribution of Risk and Resilience Factors to Suicidality Among Mental Health Seeking Adolescent Outpatients: A 3 Cross-Sectional Study. The paper addressed a topic of significant importance by examining the relationship between suicidality, peer-victimization, and resilience in youth in a psychiatric outpatient setting. Overall, the manuscript was clear and concise. The introduction included adequate literature that led to specific research questions. The measures and methods were described and replicable, and the Tables and Figures provided a good display of the results. The Discussion reviewed the results and provided convergence with the literature, limitations, and recommendations for future research.

              Response: We thank the Reviewer for appreciating our work.

Comment:

Two areas that could be improved include:

1) adding information on the internal consistency of the scales for the resilience measure;

Response: We have added Cronbach’s alpha value (=0.76) to the description of resilience measures L142-150.

2) in the first paragraph of the Discussion, much of the convergence with the literature appears to be limited to the work of co-authors and is missing other relevant studies that are applicable to the findings. I would suggest expanding this section to include other research that can aid in the clinical implications of the findings as well as guiding future experimental research.

Response: We thank the Reviewer for this comment. In the discussion, we begin by discussing this study in the context of our previous works. This is because we have data on resilience factors using the same scale as we used in the current work. In the revised manuscript, we have elaborated the discussion substantially and now relate our findings to other relevant studies and discuss their implications for our work to better contextualize this study, L335-346.

The following references were added:

1) Bessette, K.L.; Burkhouse, K.L.; Langenecker, S.A. An Interactive Developmental Neuroscience Perspective on Adolescent Resilience to Familial Depression. JAMA Psychiatry 2018, 75, 503–504.

2)  Malhi, G.S.; Das, P.; Bell, E.; Mattingly, G.; Mannie, Z. Modelling Resilience in Adolescence and Adversity: A Novel Framework to Inform Research and Practice. Transl. Psychiatry, 2019, 9(1), 316. https://doi.org/10.1038/s41398-019-0651-y.

3)  Chmitorz, A.; Kunzler, A.; Helmreich, I.; Tüscher, O.; Kalisch, R.; Kubiak, T.; Wessa, M.; Lieb, K. Intervention Studies to Foster Resilience - A Systematic Review and Proposal for a Resilience Framework in Future Intervention Studies. Clin Psychol Rev. 2018, 59, 78–100. https://doi.org/10.1016/j.cpr.2017.11.002

Reviewer 2 Report

The main purpose of the present study entitled “Contribution of Risk and Resilience Factors to Suicidality Among Mental Health Seeking Adolescent Outpatients: A Cross-Sectional Study” was to investigate associations of resilience factors with suicidality in a clinical adolescent population of a psychiatric outpatient clinic in Israel. Therefore, the authors recruited 127 participants with a final sample size of 104 outpatient mental health help-seeking adolescents who answered to self-report questionnaires on their first outpatient visit, including the Ask Suicide-Screening Questions, a battery of risk (peer-victimization and negative life events) and resilience (self-reliance, emotion regulation, close relationships and neighbourhood) measures.

Although the manuscript is interesting, it presents some incongruities. Therefore, to improve its quality specific corrections are needed.

Overall comments:

1.       The English grammar and orthography should be double-checked.

2.       An in-depth review is needed for means of resilience and suicide prevention strategies. In particular, physical activity- and exercise-related literature (e.g., sport-related studies and systematic literature reviews) is suggested.

3.       Findings should be contextualised to the specific regional setting.

4.       The conclusion section can be widely improved.

Specific comments:

Abstract

L33-36: the lack of moderating effects of resilience factors could appear in contrast to the general conclusion that study provides evidence for the protective association of resilience factors and suicidality in a psychiatric outpatient population. Please improve this passage.

Introduction

L41-42: Authors introduce the paper stating that “suicide is the second leading cause of death among adolescents in the United States”. However, the cited paper (World Health Organization,  Suicide worldwide in 2019) reports: “Suicide was the fourth leading cause of death in 15–19-year-olds for both sexes, with the number of deaths relatively similar between females and males in this age group”. Please emend or refer to more detailed sources (see: https://www.who.int/data/gho/data/themes/mortality-and-global-health-estimates/ghe-leading-causes-of-death).

L48: the association […] seems (with s). Please emend.

L51-52: literature reports that physical activity, exercise and sport present significant and well-established effects on many mediators of the association between peer-victimization and suicidality, such as depression, self-esteem, anxiety, emotion regulation, and cognitive aspects. However, surprisingly the paper completely overlook these potential means of resilience and suicide prevention strategies. Authors may refer to Biddle, S. J., Ciaccioni, S., Thomas, G., & Vergeer, I. (2019). Physical activity and mental health in children and adolescents: An updated review of reviews and an analysis of causality. Psychology of Sport and Exercise, 42, 146-155 and Vancampfort, D., Hallgren, M., Firth, J., Rosenbaum, S., Schuch, F. B., Mugisha, J., ... & Stubbs, B. (2018). Physical activity and suicidal ideation: A systematic review and meta-analysis. Journal of affective disorders, 225, 438-448.

P62: please improve the section on suicide prevention strategies. Right now it appears just sketched.

Methods

L79-91: to evaluate whether the study had enough participants to detect an association, authors should explain what are the reasons for recruiting the number of patients included and analysed. Did authors determined a statistical power analysis of the study?

To disclose potential for bias, authors should explain whether the persons assessing the outcomes for the study were “blinded” or “masked” to the exposure status of the patients.

L116: positive with one “s”. Please emend.

L116: current with two “r”. Please emend.

L117: “imidiately” should be immediately. Please emend.

L118: “intervantional” should be interventional. Please emend. Reviewers should not correct similar errors from such high-level scholars.

L128: team psychiatrist or psychiatrist team?

L148-152: please provide a question example for the “Patient Health Questionnaire-9”. Also there is no need to repeat the word questionnaire since it is already stated in its name.

L153-157: please provide a question example for the Generalized Anxiety Disorder 7 questionnaire.

Statistical Analysis

Please indicate the R Stats Package version.

Results

L200: to avoid repetitions, youth can be substituted with “adolescents”, “participants”, “individuals” or similar.

Tables

Table 1: please explain on a note the meaning of letters and acronyms (n, N, SD, GAD-7, PHQ-9). To be consistent, authors could always use the lower case “n” and substitute “M” (see: Risk factors, M) with “mean”.

Table 2 and 3: similarly, please include in a note the meaning of acronyms.

Discussion

Although findings are well discussed, strategies of suicide prevention should be better examined both in clinical and general young populations. Moreover, retrieved results should be contextualised to the specific regional setting of Israel providing meaningful comparisons with similar and different contexts.

Finally, the end of discussion section needs more precise “future of research” proposals.

Conclusions

This section appears too generic and should be improved explaining to the reader the importance of this study elaborating on the significance of the findings.

References

Please double-check the correctness of all references. Different formats seem to be used (e.g., for the journal title abbreviations and punctuation).

L408-412: not sure whether a paper that has to be submitted and it has not peer-reviewed could be cited.

Author Response

Reviewer #2
Abstract

Comment:

L33-36: the lack of moderating effects of resilience factors could appear in contrast to the general conclusion that study provides evidence for the protective association of resilience factors and suicidality in a psychiatric outpatient population. Please improve this passage.

Response: Thank you for this comment; we changed " Resilience factors did not moderate…"  to “Nevertheless, high peer victimization was found to be associated with a greater chance of suicidality across all levels of resilience." We also addressed this issue in the discussion starting in L32-34.

Comment:

L41-42: Authors introduce the paper stating that “suicide is the second leading cause of death among adolescents in the United States”. However, the cited paper (World Health Organization, Suicide worldwide in 2019) reports: “Suicide was the fourth leading cause of death in 15–19-year-olds for both sexes, with the number of deaths relatively similar between females and males in this age group”. Please emend or refer to more detailed sources (see: https://www.who.int/data/gho/data/themes/mortality-and-global-health-estimates/ghe-leading-causes-of-death).

Response: We replaced the reference with data from the NIMH website and changed the introduction accordingly in L43-45.

Comment:

L51-52: literature reports that physical activity, exercise and sport present significant and well-established effects on many mediators of the association between peer-victimization and suicidality, such as depression, self-esteem, anxiety, emotion regulation, and cognitive aspects. However, surprisingly the paper completely overlook these potential means of resilience and suicide prevention strategies. Authors may refer to Biddle, S. J., Ciaccioni, S., Thomas, G., & Vergeer, I. (2019). Physical activity and mental health in children and adolescents: An updated review of reviews and an analysis of causality. Psychology of Sport and Exercise, 42, 146-155 and Vancampfort, D., Hallgren, M., Firth, J., Rosenbaum, S., Schuch, F. B., Mugisha, J., ... & Stubbs, B. (2018). Physical activity and suicidal ideation: A systematic review and meta-analysis. Journal of affective disorders, 225, 438-448.

Response:  We thank the Reviewer for this important comment. Indeed, physical activity is an established protective factor in peer-victimized youth. However, our risk and resilience battery did not collect data on lifestyle factors, and therefore we do not have data on physical activity in our study population.

In the revised manuscript, we have now added this as a limitation of our study. We have added the suggested references, as well as a discussion of the role of physical activities in future suicide prevention and intervention studies, L369-374.

Comment:

P62: please improve the section on suicide prevention strategies. Right now it appears just sketched.

Response:  We have added two paragraphs to discuss potential interventions promoting resilience (Page 9, 3rd and 4th paragraph, L320-346). The following references were added:

1) Mayoral, M.; Valencia, F.; Calvo, A.; Roldan, L.; Espliego, A.; Rodriguez-Toscano, E.; Kehrmann, L.; Arango, C.; Delgado, C. Development of an Early Intervention Programme for Adolescents with Emotion Dysregulation and their Families: Actions for the Treatment of Adolescent Personality (ATraPA). Early Interv Psychiatry 2020, 14(5), 619-624. doi: 10.1111/eip.12934. Epub 2020 Feb 5. PMID: 32026614.

2)  Dekel, I.; Hertz-Palmor, N,; Dorman-Ilan, S.; Reich-Dvori, M.; Gothelf, D., Pessach, I.M. Bridging the Gap between the Emergency Department and Outpatient Care: Feasibility of a Short-Term Psychiatric Crisis Intervention for Children and Adolescents. Eur Child Adolesc Psychiatry  2021, 26, 1–7. doi: 10.1007/s00787-021-01896-2.

3)  Klomek A.B.; Catalan L.H.; Apter A. Ultra-brief Crisis Interpersonal Psychotherapy Based Intervention for Suicidal Children and Adolescents. World J Psychiatry 2021, 11(8), 403-411. doi: 10.5498/wjp.v11.i8.403. PMID: 34513604; PMCID: PMC8394689.

4) Malhi, G.S.; Das, P.; Bell, E.; Mattingly, G.; Mannie, Z. Modelling Resilience in Adolescence and Adversity: A Novel Framework to Inform Research and Practice. Transl. Psychiatry, 2019, 9(1), 316. https://doi.org/10.1038/s41398-019-0651-y.

5) Chmitorz, A.; Kunzler, A.; Helmreich, I.; Tüscher, O.; Kalisch, R.; Kubiak, T.; Wessa, M.; Lieb, K. Intervention Studies to Foster Resilience - A Systematic Review and Proposal for a Resilience Framework in Future Intervention Studies. Clin Psychol Rev. 2018, 59, 78–100. https://doi.org/10.1016/j.cpr.2017.11.002

Methods
Comment:
L79-91: to evaluate whether the study had enough participants to detect an association, authors should explain what are the reasons for recruiting the number of patients included and analysed. Did authors determined a statistical power analysis of the study?

Response:  We conducted a power analysis before recruitment and have now added this to the revised manuscript. L87-91

“A power analysis using G*Power 3.1.9.4 (25) showed that with strong hypothesized effect sizes of odds ratio>2 (or 0.48 for protective effect), a sample of N=100 would be required to observe effects with an α criterion <.05 with 80% statistical power.”

Comment:

To disclose potential for bias, authors should explain whether the persons assessing the outcomes for the study were “blinded” or “masked” to the exposure status of the patients.

Response:
We apologize for the lack of clarity. In the revised manuscript, we elaborate that outcomes were assessed through self-administered questionnaires, and in cases where patients required assistance team members were blinded to their responses in L101-105.

“The questionnaires were self-administered, except for cases where patients experienced difficulties in reading or understanding of the questions, in which case a member of the research team would read items out loud and provide instructions. Team members were not aware of the patient’s responses, as they only read the questions and enabled patients to click on the relevant answer independently.”

Comment:

Reviewers should not correct similar errors from such high-level scholars.

Response: We sincerely apologize for these typos. We have corrected them in the revised version.

Comment:

L128: team psychiatrist or psychiatrist team?

Response: We corrected the ambiguous description to “a psychiatrist”.

Comment:

L148-152: please provide a question example for the “Patient Health Questionnaire-9”. Also there is no need to repeat the word questionnaire since it is already stated in its name.
L153-157: please provide a question example for the Generalized Anxiety Disorder 7 questionnaire.

Response: We now provide an example item for each of these questionnaires, please see below.

For the PHQ-9: “Feeling down, depressed, or hopeless”

For the GAD-7: “Feeling nervous, anxious or on edge”

Statistical Analysis
Comment:
Please indicate the R Stats Package version.

Response: We have now indicated the stats package version (page 5, paragraph 1, L209):

“Analyses were conducted with the ‘stats’ package (version 4.0.3) in R”

Results

Comment:
L200: to avoid repetitions, youth can be substituted with “adolescents”, “participants”, “individuals” or similar.

Response: We have edited the section in line with the Reviewer’s suggestion.

Tables

Comment:
Table 1: please explain on a note the meaning of letters and acronyms (n, N, SD, GAD-7, PHQ-9). To be consistent, authors could always use the lower case “n” and substitute “M” (see: Risk factors, M) with “mean”. 

Response: We thank the reviewer for this important note. We have clarified the abbreviations used in the caption. As elaborated in the caption, uppercase N and lowercase n refer to the common distinction between the entire sample (N) and a subsample of participants (n), such as suicidal vs. non-suicidal groups. We elaborated on that in the caption.

Comment:
Table 2 and 3: similarly, please include in a note the meaning of acronyms.

Response: We added a similar caption to tables 2+3.

Discussion

Comment:

Although findings are well discussed, strategies of suicide prevention should be better examined both in clinical and general young populations. Moreover, retrieved results should be contextualized to the specific regional setting of Israel providing meaningful comparisons with similar and different contexts. Finally, the end of discussion section needs more precise “future of research” proposals.

Response: As mentioned earlier, we added two paragraphs to discuss future research directions in the context of this and previous studies. We specifically elaborated on how our work can inform future interventions to enhance resilience (Page 9, paragraphs 2 and 3, L320-346).

Conclusions

Comment:

This section appears too generic and should be improved explaining to the reader the importance of this study elaborating on the significance of the findings.

Response: We have rewritten the conclusion section focusing on the clinical implications of our findings to resilience research in the context of adolescent suicidality risk.